# Queueing theory model of mTOR complexes' impact on Akt-mediated adipocytes response to insulin

Sylwester M. Kloska[1,2]*, Krzysztof Pałczyński[2], Tomasz Marciniak[2], Tomasz Talaśka[2], Marissa Miller[3], Beata J. Wysocki[4], Paul H. Davis[4], Ghada A. Soliman[5], Tadeusz A. Wysocki[2,3]

1 Department of Forensic Medicine, Nicolaus Copernicus University Ludwik Rydygier Collegium Medicum, Bydgoszcz, Poland, 2 Faculty of Telecommunications, Computer Science and Electrical Engineering, Bydgoszcz University of Science and Technology, Bydgoszcz, Poland, 3 Department of Electrical and Computer Engineering, University of Nebraska-Lincoln, Omaha, Nebraska, United States of America, 4 Department of Biology, University of Nebraska at Omaha, Omaha, Nebraska, United States of America, 5 Department of Environmental, Occupational, and Geospatial Health Sciences, City University of New York, Graduate School of Public Health and Healthy Policy, New York, NY, United States of America

* 503013@stud.umk.pl

**Data Availability Statement:** All relevant data are within the paper and its Supporting Information files. The source code of the described model can

## Abstract

A queueing theory based model of mTOR complexes impact on Akt-mediated cell response to insulin is presented in this paper. The model includes several aspects including the effect of insulin on the transport of glucose from the blood into the adipocytes with the participation of GLUT4, and the role of the GAPDH enzyme as a regulator of mTORC1 activity. A genetic algorithm was used to optimize the model parameters. It can be observed that mTORC1 activity is related to the amount of GLUT4 involved in glucose transport. The results show the relationship between the amount of GAPDH in the cell and mTORC1 activity. Moreover, obtained results suggest that mTORC1 inhibitors may be an effective agent in the fight against type 2 diabetes. However, these results are based on theoretical knowledge and appropriate experimental tests should be performed before making firm conclusions.

## Introduction

### Biological importance

A key hormone that controls blood glucose levels is insulin. This hormone is secreted by the β-cells of pancreatic islets. Insulin facilitates glucose uptake in peripheral tissues including the muscle, and adipose tissue [1]. It inhibits glucose production from non-glucose sources by inhibiting gluconeogenesis and glycogenolysis, while stimulating glycogen synthesis. The hormone with the opposite effect of insulin is glucagon [2]. Both of these hormones together are primarily responsible for the maintenance of glucose homeostasis in mammals.

The attachment of insulin to the insulin receptor starts a cascade of reactions responsible for the absorption of glucose inside the cell [3]. One of the main effects of this cascade is the translocation of glucose transporter 4 (GLUT4) from the center of the cell towards the cell

be accessed by: https://doi.org/10.5281/zenodo.
7117138.

**Funding:** This work was supported by the National
Science Center (Narodowe Centrum Nauki, NCN) of
Poland (https://www.ncn.gov.pl/) in terms of Opus-
17 Program [2019/33/B/ST6/00875 awarded to
TAW]. The funders had no role in study design,
data collection and analysis, decision to publish, or
preparation of the manuscript.

**Competing interests:** The authors have declared
that no competing interests exist.

membrane. GLUT4 is a protein that facilitates the diffusion of glucose along a concentration gradient–from a higher concentration in the blood to a lower concentration in the cell. The participation of GLUT4 in the transport of glucose inside the cell increases the amount of transported glucose molecules by 30 times [4, 5].

Adequate management of glucose levels in the cell is crucial to maintain a healthy environ-ment in the cell and its function. One of the mechanisms that supervise the maintenance of adequate blood glucose levels is through mammalian target of rapamycin (mTOR) kinase. mTOR links with other proteins and forms two protein complexes described as mTORC1 and mTORC2. These complexes are responsible for regulation of various important processes inside the cell, including cell growth regulation, cell proliferation, cell motility, cell survival, protein synthesis, autophagy, DNA transcription, and metabolism [6]. The dysregulation and incorrect activity of mTOR complexes can lead to diseases such as obesity, diabetes and even cancer [7, 8]. One of the proteins that regulate the mTORC1 complex is the Rheb protein [9, 10]. It is one of the key mTORC1 activating proteins. However, one enzyme in the glycolytic pathway–glyceraldehyde 3-phosphate dehydrogenase (GAPDH), has a high affinity for the Rheb protein [11]. When GAPDH enzyme molecules are not involved in the reaction that pro-duces 1,3-bisphosphoglycerate (1,3-BPG) from glyceraldehyde 3-phosphate (G3P), they com-bine with Rheb protein molecules, depriving the mTORC1 complex its key activator, leading to inactivity of the mTORC1. When the cell has normal/high concentrations of G3P, GAPDH molecules are busy processing G3P, so Rheb can freely bind to mTORC1 and activate it. Depending on the above-described mTORC1 activation process, the amount of GLUT4 parti-cles varies. For this reason, we decided to prepare a computational model capable of predicting the number of active GLUT4 particles that are capable of participating in glucose transport.

## Queueing theory

Typically, cellular signaling networks have been modeled using a set of ordinary differential equations (ODEs) [12]. Using these equations, it is possible to demonstrate the changes that occur in the cell during rest and in response to external stimuli causing upstream signals. However, when using ODEs, the fluctuations in the cell leading to local changes (e.g., tempera-ture) are not taken into account, which influences the values of the kinetic constants that affect the way the cell responds. To map the intracellular environment more accurately, as well as the random variation, a model based on the queueing theory can be useful. Queueing theory was mainly used in telecommunications and engineering [13–16]. Additionally, it is suitable for modeling stochastic processes in cells. The idea to use a method commonly used in telecom-munications comes from the fact that signaling paths, similar to the transmission of internet packets, transmit information from node to node. Likewise, in a cell, signaling molecules are passed on, activating subsequent elements (proteins) of the cascade. To date, the queueing the-ory approach has been used to model simple metabolic networks [17], metabolic pathways such as glycolysis [18] and the Krebs cycle [19]. The presented model is an extension of the work [20] to include loops related to the regulation of cellular metabolism by mTOR com-plexes and mTORC1 regulation via GAPDH availability, or more precisely–'occupancy'. In the case of models such as the one presented here, which use a large number of variables, the application of the queuing theory seems to be more optimal than the use of ODEs. The model is capable of achieving stability. Another advantage of using queueing theory to model signal-ing pathways is that they require significantly less computing power compared to ODE mod-els. For this reason, simulation can be carried out practically in real time. Due to the short duration of the simulation, it can be used to learn about the relationships caused by manipula-tions of specific kinetic constants or concentrations, which also has its advantages when

considering the reactions that are not well studied/established. To confirm the correctness of the obtained simulation results, the simulated data were compared with the results of laboratory experiments [21]. Finally, using the queueing theory gives the possibility of expanding the model with further reactions, without major interference in the course of the previously described, due to the fact that they are based on empirically obtained values. Therefore, the model is adapted to be supplemented with the development of the state of knowledge about the given signaling or metabolic pathways. Moreover, it can be used to theoretically test the kinetic changes brought about by potential mTORC inhibitors [22].

The aim of this work is to present a comprehensive model of cellular response to insulin, which leads to GLUT4 translocation and mTORC activation as a part of processes responsible for maintaining proper cellular glucose concentration. Fig 1 shows the links between molecules involved in the insulin signaling pathway. The research hypothesis of this work is the ability to simulate the cellular response to insulin and track changes in the concentrations of proteins involved in this response using queueing theory based simulation model. The presented model shows the mechanism of mTORC1 influence on mobilization of GLUT4

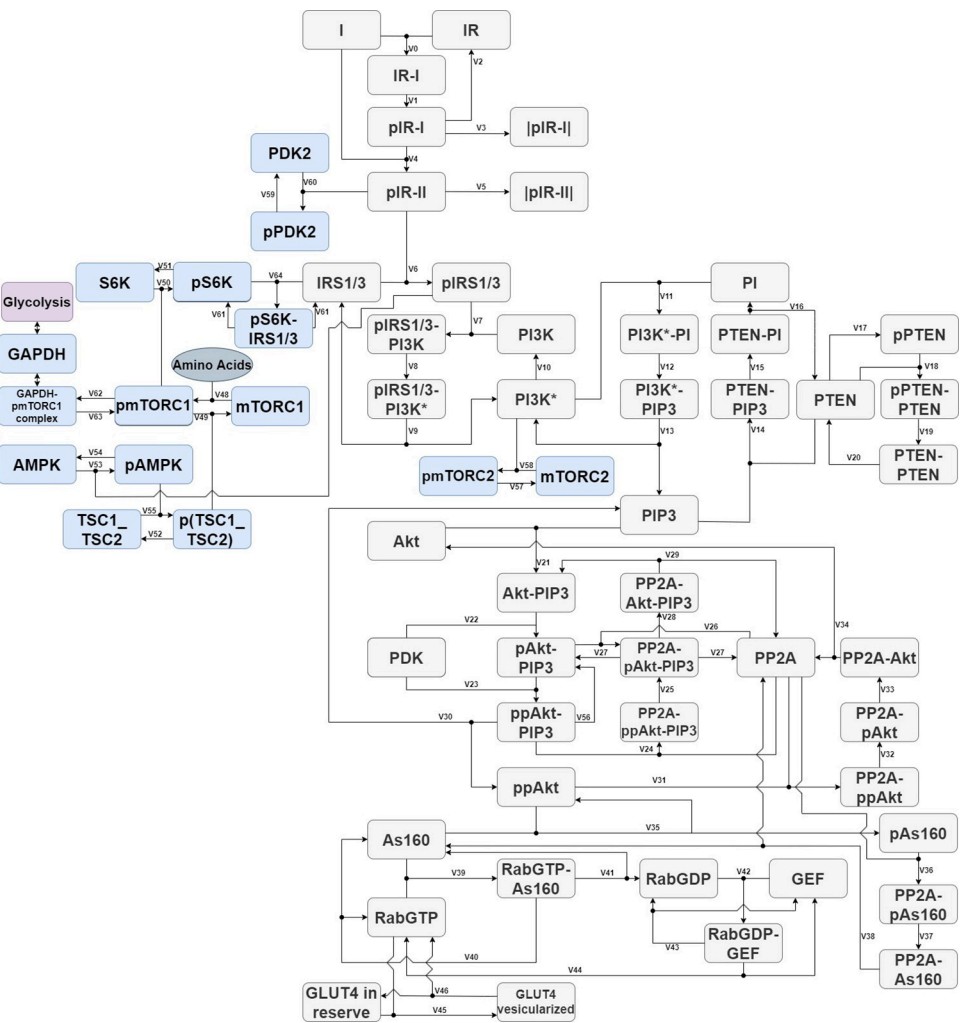

**Fig 1. Diagram illustrating the computational model of the insulin signaling pathway.** The cascade of reactions begins when insulin binds to the insulin receptor. The scheme includes the most important proteins in the PI3K/Akt/ mTOR pathway which play a role in the cellular response to insulin.

particles. Since mTORC1 has been reported in literature as having an impact on glucose uptake [23, 24].

The presented model is an extension of previously described simulation of insulin mediated GLUT4 translocation [20]. Since then the mTORC1 signaling pathway connections with Akt-mediated insulin response has been described [25, 26]. This work presents a model where those connections have been included. Moreover, the paper studies regulation of mTORC1 activity by the glycolytic enzyme GAPDH, which has high affinity for the mTOR activator protein–Rheb. To train the model we have used genetic algorithm (GA) to optimize the kinetic coefficients. The achieved results allow to conclude that artificial intelligence (AI) algorithms, in this case the genetic algorithm, can be effective tools for optimizing computational models. In order to validate the obtained results, we present multiple variants of mTORC1 activity that can be practically obtained through the administration of an mTORC1 inhibitor, such as rapamycin [27].

## Methodology

The endpoints of the Akt-mediated insulin signaling pathway are well characterized [28–30]. Therefore, by comparing the experimental and computational results, it can be assessed whether the model works properly. The values of kinetic constants and concentrations of signaling molecules were obtained by searching the PubMed database. Simulations were performed separately for 50 independent cells, which mimic human adipocytes. This type of cell was chosen because of the availability of literature data, which was used in the development of the model. For each of the cells, the concentrations of all molecules participating in the signaling pathway were randomly chosen from the given range, limited by 10% Gaussian noise. According to the queueing theory, the current concentrations of individual molecules in each cell are separate 'stores'–queues [18, 31]. The speed of the response determines the probability of passing from one queue to the next. The simulation results are averaged over the entire cell population. A network based on queues can be used to model reactions whose rates change dynamically and randomly. The simulation was performed in C# 8.0. All the results were obtained using 1ms time increments; however, the simulation allows the choice of any user-selected time increment. While changing the time increment, one should pay attention to the fact that the probabilities of the reaction occurrence are <1. Detailed information on the equations, kinetic constants, and initial concentrations can be found in the S1 File.

The Genetic Algorithm [32, 33] was used to tune the model of interconnected queues realizing Michaelis-Menten equations. Each 'chromosome' consisted of linear coefficients for selected group of queues scaling their probability of reaction occurrence. The population of GA consists of ten 'chromosomes'. In each epoch, every 'chromosome' is evaluated and the two 'chromosomes' with the best scores are chosen. The process of 'chromosome' evaluation consists of performing three simulations with a set of kinetic constants, linear coefficients stored in each 'chromosome', and a value of available GAPDH. Each simulation used a different value of available GAPDH taken from a set {0%, 20%, 50%, 100%}. One simulation was formed emulating 50 cells working in parallel to each other. The evaluation step was added to measure, 1) how many cells reached the maximum value of GLUT4 in vesicles for available GAPDH equal to 100%, 2) how many cells reached the minimum value of GLUT4 in vesicles for available GAPDH equal to 0%, and 3) how distant is the number of cells that reached the maximum value of GLUT4 in vesicles for available GAPDH equal to 50% from aforementioned results for GAPDH equal to 100% and 0%.

To validate the model, theoretical inhibition of mTORC1 was used to test the effects of changes in reduction of its activity. One of the inhibitors of mTOR complexes' activity is

rapamycin. Previous studies show that rapamycin causes a number of side effects, including increased risk of infection [34], increased incidence of cancer [35], weight disorders, hyperlipidemia, and diabetes-like metabolic disorders [36]. For this reason, it seems necessary to develop drugs that selectively affect mTORC1 activity, while at the same time not having such significant side effects, like astragaloside IV (As-IV) [37]. As-IV was proven to be effective mTORC1 inhibitor and reduced mTORC1 signaling in mice. The data obtained from the presented model can be used in the study of the kinetics of reactions in the insulin signaling pathway, which will help to select the appropriate place where the influence of therapeutics could have the best effect.

Without insulin activating the cascade and mobilizing GLUT4 to move towards the cell membrane, there are approximately 18,200 GLUT4 molecules proximate to the cell membrane [38], ready to transport glucose inside the cell. This number increases to approximately 195,000 as a result of insulin-stimulated activation [38, 39]. However, these are not total GLUT4 stocks. In fact, the cell has a large reservoir that it can use in extreme cases. The said number 195,000 accounts for approximately 50% of total GLUT4 [21].

To illustrate the changes caused by the influence of GAPDH molecules on mTORC1 activation, two varying scenarios are described below (Fig 2). These scenarios focus on different cellular conditions such as glucose levels and the intensity of glycolysis.

Scenario I–the concentration of glucose in the blood is elevated after eating and the insulin signaling pathway works correctly. As a result, GLUT4 molecules are mobilized to migrate to the cell membrane, where they facilitate the flow of glucose from the blood to inside the cell. The glucose level in the blood drops, while the cellular level of glucose rises. To avoid the situation where glucose molecules leave the cell, glucose is phosphorylated and becomes G6P. There are two destinations for G6P molecules: 1) the glycolytic pathway or 2) glycogenesis, the formation of glycogen.

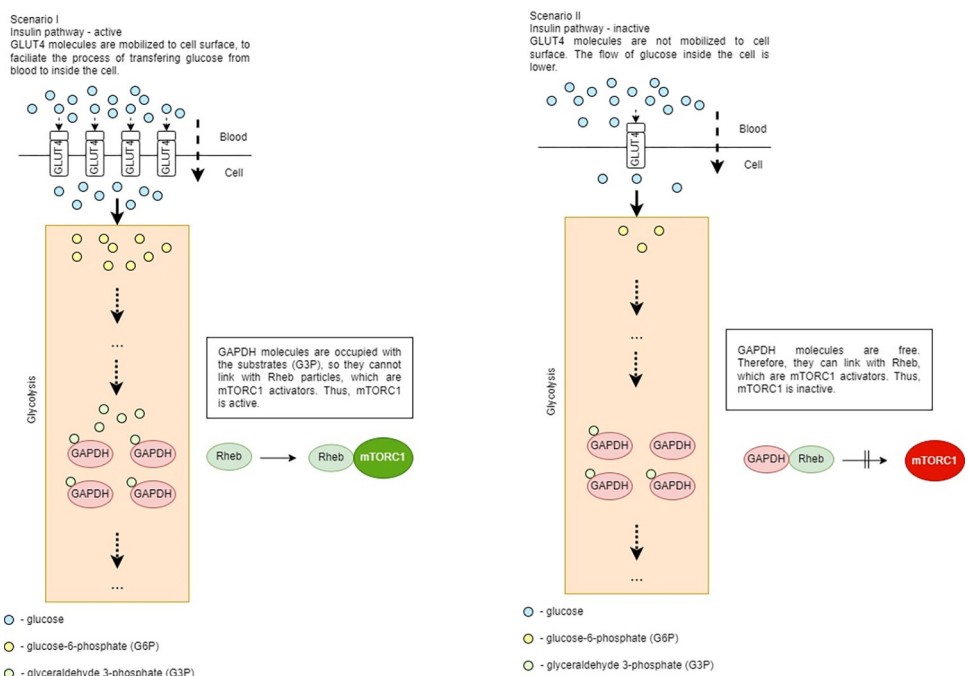

**Fig 2. Various scenarios of mTORC1 activity depending on GAPDH 'occupancy'.**

When G6P enters glycolysis, the sequence of reactions takes place and glyceraldehyde 3-phosphate (G3P) molecules are formed. G3P is converted into 1,3-bisphosphoglycerate (1,3BPG) by GAPDH.

GAPDH is particularly important because it is involved in regulation of mTORC1 activity. GAPDH concentration levels in the cell do not change drastically rather they oscillate around the same values. However, what changes is their state–they can be either 'occupied' with processing G3P molecules, or if there are more enzyme molecules than substrate molecules, the excessive amount of enzyme molecules is free. Those free GAPDH molecules connect with Rheb protein and activate mTORC1. It remains unknown how Rheb stimulates the activity of mTORC1.

Scenario II–the organism is in state of prolonged fasting causing a decrease in the supply of extracellular glucose and ceasing insulin secretion. Without the release of insulin from the blood, the reaction remains inactivated and GLUT4 remains stationary and unable to transport glucose. In this situation, the stored amounts of glycogen are hydrolyzed and the basic levels of G6P are maintained. As previously described, glycolysis runs as normal. However, the amount of formed G3P molecules is lower than in Scenario I. In fact, there is larger amount of GAPDH molecules than G3P molecules. Therefore, the free GAPDH molecules can freely bind with Rheb protein, resulting in mTORC1 inactivation.

To conclude, increased extracellular supply of glucose activates insulin signaling. The glycolytic flux is increased and the GAPDH molecules are occupied with processing G3P molecules. As a result, Rheb molecules are floating freely and can bind to and activate mTORC1.

However, the conditions presented in both scenarios are extreme and practically unrealistic in the cell, as the probability of such extreme conditions as 0 or 100% 'occupancy' of GAPDH is low. In a cell, most often intermediate conditions prevail.

## Results

### Effect of GAPDH and mTORC1 on the amount of GLUT4 involved in glucose transport

A working, stable queueing theory-based model of the insulin signaling pathway was obtained. The presented study was aimed at illustrating the interrelationships between the levels of GLUT4, GAPDH, and mTORC1. These relationships have a significant impact on how the cell responds to insulin and extracellular glucose supply. The results obtained with the use of the model are consistent with the current state of knowledge [10, 40]. The amount of GLUT4 particles ready to take part in the glucose transport process is significantly dependent on the amount of 'occupied' GAPDH. When the system is not inhibited, less than 200,000 GLUT4 molecules are involved in the transport of glucose to the cell. However, depending on the level of activity that is influenced by both GAPDH and indirectly by mTORC1, this number fluctuates. Fig 3 shows the relationship between the level of GLUT4 in the vicinity of the cell membrane and the level of 'occupied' GAPDH. Depending on the condition of the cell, as well as mTORC1 activity, the amount of GLUT4 mobilized can vary considerably (Figs 4 and 5). The greater amount of GAPDH involved in substrate processing allows Rheb to link freely with mTORC1. mTORC1 activity and GLUT4 level are correlated with each other [41, 42]. The same conclusions can be drawn by analyzing the obtained results on the charts.

We also tested the effect of lowering mTORC1 activity, e.g., through the use of drugs, on the amount of GLUT4 particles, while assuming different levels of GAPDH 'occupancy' (Fig 4). Analogous studies were performed for different levels of GAPDH with respect to mTORC1 activity (Fig 5). Both mTORC1 activity and the amount of 'occupied' GAPDH significantly influences the amount of GLUT4 and can contribute to lowering the amount of GLUT4

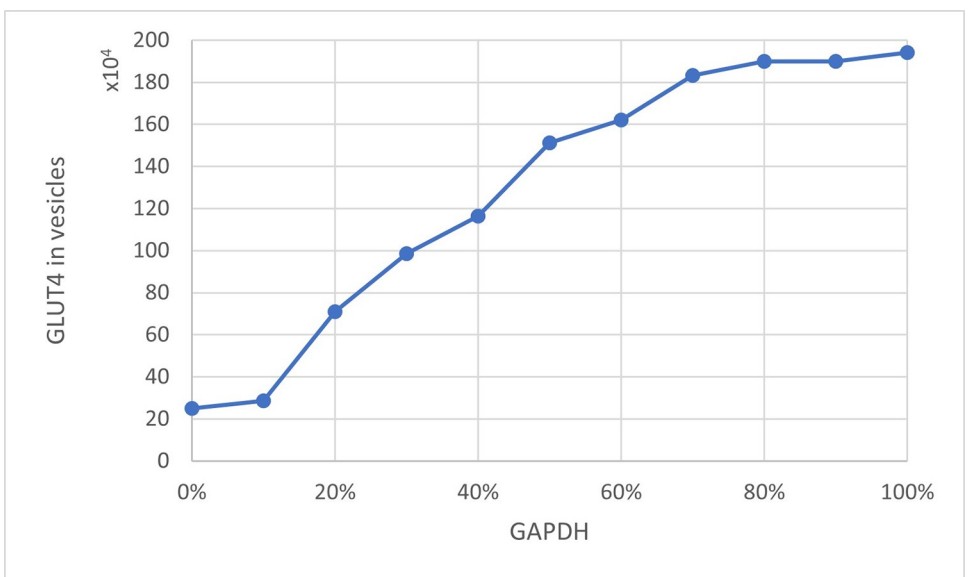

**Fig 3. The relationship between the amount of GLUT4 particles in the cell membrane area and the level of 'occupied' GAPDH.**

particles involved in glucose transport (Fig 5). The scenario in which all the GAPDH particles present in the cell are busy processing its substrate so that the mTORC1 can be fully active, keeps the amount of GLUT4 in vesicles at the maximum level (Fig 5). The presented results indicate that drugs that can significantly decrease mTORC1 activity (at least 50% inhibition) are of great importance for the amount of GLUT4 particles directed to the cell membrane for glucose transport inside the cell. Similar conclusions can be drawn from the results presented

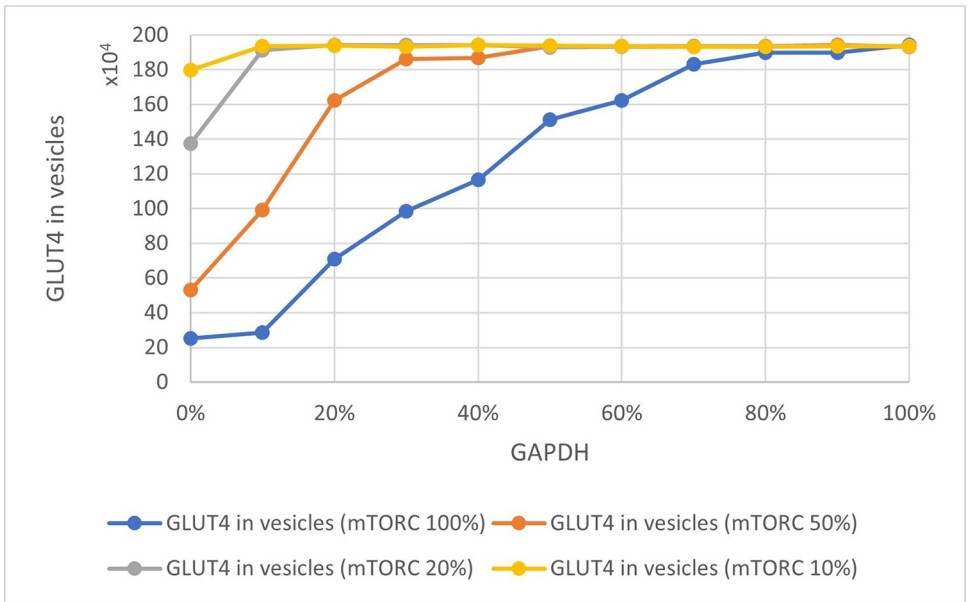

**Fig 4. Relationship of GLUT4 in vesicles and 'occupied' GAPDH.** Colored lines indicate different levels of mTORC1 activity.

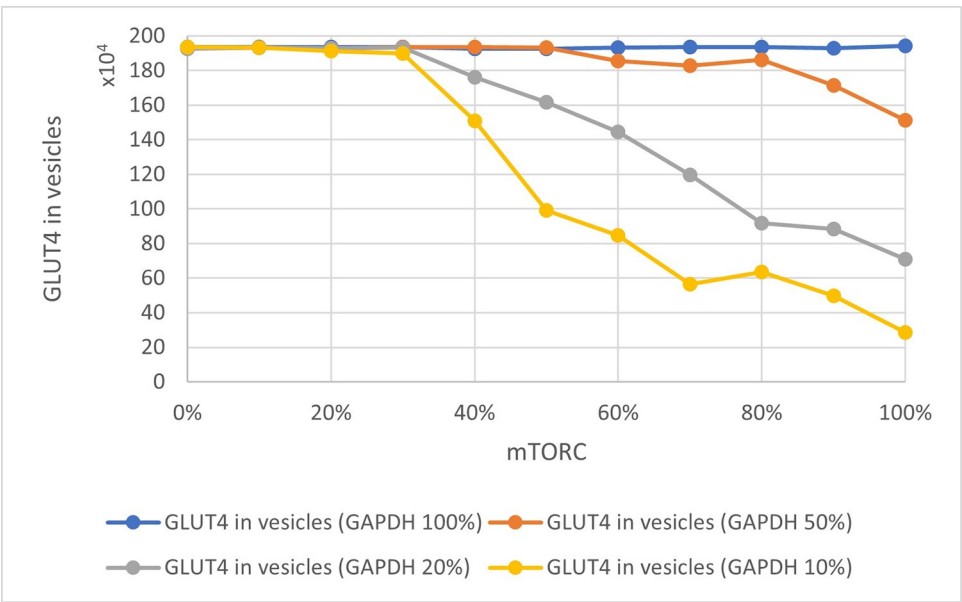

**Fig 5. Dependence of GLUT4 level in vesicles in relation to mTORC1 activity.** The colored lines indicate the different levels of 'occupied' GAPDH.

by Rajan et al. [43] and Veilleux et al. [44] which confirms the validity of the method we presented.

## Discussion

### Identification of key nodes in insulin signaling

Practical application of the conclusions of the described scenarios for GAPDH and mTORC1 allowed the identification of key nodes for the appropriate cell response to insulin, and confirmed previous experimental results described in [45]. In [45] the authors explained and proved the important role of S6 kinase (S6K). mTORC1 participates in its phosphorylation. S6K is crucial because it is the link between the mTORC1 loop and the rest of the proteins responsible for insulin signaling. Signaling between mTORC1 and S6K causes a negative-feedback loop which lowers cellular sensitivity for insulin. The activation of the mTORC1/S6K loop leads to increased degradation of insulin receptor substrate 1/3 (IRS1/3) and therefore influences the amount of GLUT4 in vesicles. This entire process affects how many glucose molecules enter the cell from the bloodstream.

The experimental results, as well as those obtained in the presented model, indicate that the insulin response system is very complex and depends on many elements that regulate it. It is characterized by high instability, small changes that can lead to a greatly altered cell response, causing disease such as type 2 diabetes, where the cells become insensitive to insulin. As shown in the above model, there are many elements that can cause glucose malabsorption.

Sonntag et al. [46] focused on determining which of the 'nodes' of the insulin signaling pathway influences AMP-activated protein kinase (AMPK) activity. The equations described in the [46] are based on the mass action law. The obtained results state that IRS1/3 is the 'node' influencing AMPK. The model proposed by Sonntag et al. focused on simplifying the insulin signaling pathway and it does not take into account several 'nodes' that play a significant role in this process. Therefore, the combination of the data and results presented by [46] was a

valuable source in the preparation of the model based on the queueing theory. GA was used to find an appropriate scaling of the values so that the model as a whole would work properly.

The presented model has several limitations. It does not take into account other signaling pathways or individual reactions that are also connected to and influence signaling proteins. This is especially true for the Akt protein, which is the central node in the presented signaling model. Moreover, a model based on literature data will only be as good as the available data. However, we do not question the reliability of other research teams and their published results. Another of the limitations is that in queueing theory, each simulation gives one realization of the stochastic process, while ODE gives an averaged solution. Therefore, a limitation is that depending on the number of cells for which one runs simulations and then averages them, this is how accurate the result will be. Therefore, the model presented here is for averaged results for 50 cells.

## mTORC1 activity and related treatment strategies

The results of the described model could be used as a suggestion in the process of developing new drugs, including drugs that increase insulin-sensitivity in peripheral tissues such as the muscle and adipose tissue (e.g., Metformin). Identifying key 'nodes' throughout the signaling pathway could guide researchers in helping cells regain their original insulin sensitivity. However, due to the complexity of connections between all signaling molecules, this task is very difficult.

mTORC1 plays an important role in the maintenance of an adequate level of glucose in the blood. When necessary, i.e., in a nourished state, mTORC1 activity stimulates pancreatic β-cells to secret insulin, thus maintaining adequate glucose tolerance. However, studies in mice [47, 48] show that mTORC1 overactivity may cause a faster deterioration in β-cell function and consequently complications with glucose homeostasis. Therefore, the use of mTORC1 inhibitors to improve glucose tolerance has been considered. Previous studies in mice have shown that S6K knockdown or inhibitors that reduce S6K phosphorylation make cells more insulin sensitive [49, 50]. The results obtained with the use of the queueing theory model confirm earlier reports [45] that mTOR/S6K inhibition could be a therapeutic target in type 2 diabetes.

One of the most common prototype mTOR inhibitors is rapamycin. However, the use of rapamycin has been counterproductive, inducing insulin resistance and disrupting glucose homeostasis in the body [51]. Rapamycin is an effective inhibitor of mTORC1. Most researchers agree that rapamycin does not inhibit mTORC2 at least in the acute stimulation [52]. Few researches suggest that rapamycin inhibits mTORC2 only in some cell types and only with chronic administration due to inhibiting to mTORC2 assembly [53, 54]. Knowing the function and the importance of this complex in signaling pathway, it is no wonder that long-term mTOR inhibition interferes with the body's response to insulin. Due to the fact that rapamycin affects both mTORC1 and mTORC2, it can be concluded that it is worth testing substances that act selectively on only one of these complexes.

Research by Tao et al. [22] provided useful information on the influence of inhibitors on mTORC kinetics and activity. mTORC1 activity can be completely inhibited by ATP competitive inhibitors, like BEZ235 or PI103, while non-competitive ATP inhibitors, like rapamycin, inhibits mTORC1 activity only partially by interacting with the FRB (FKBP-rapamycin-binding) domain. By affecting kinetic properties of mTOR, they influence the process of glucose absorption in the cell. These types of results and information can provide data that can be complemented by the presented model. In this way, it will be possible to characterize changes in the entire signaling pathway induced by the use of mTORC1 inhibitors and evaluate the effect of this inhibition on the amount of GLUT4 in vesicles.

Increased mTORC1 activity has been also reported in many types of cancer [8]. mTOR is one of the factors influencing the development and growth of cells. Its excessive activity encourages cancer cells to further grow, divide and invade other healthy tissues. For this reason, it was decided to test mTORC1 inhibitors in cancer therapy [27], as they appeared to be an effective tool for coercing cancer cells into apoptosis. Although many mTORC inhibitors have been tested, some of them have been approved for therapy, however, their therapeutic capacity is relatively low. For this reason, they are most often used in combination with other anticancer drugs. In addition, their side effects must be considered. Palm et al. [55] demonstrated on mouse model of pancreatic cancer that rapamycin may even promote cell proliferation at poorly vascularized sites of the tumor. In view of all this information, it remains vital to study mTOR more thoroughly because its participation in cancer metabolism is undeniable [56], which is why it seems to be such an important research direction. The presented model can be used for this type of research, during the theoretical phase, where the likely results of their use can be determined using the data on the influence of new drugs on mTOR kinetics.

## Conclusions

A queueing theory model of mTORC1 and mTORC2 impact on Akt-mediated cell response to insulin was prepared. The presented results show that queuing theory can effectively model the manipulation of mTORC1 kinase activity influences the amount of GLUT4 used to transport glucose inside the cell, and therefore influences the concentration of glucose in the cell. The work shows suggestions of alternative targets for treating type 2 diabetes. Due to the number of people with diabetes and the existing methods of relieving symptoms, without treating the disease, any new therapeutic target may prove to be crucial. However, it should be noted that due to the nature of the studies performed, our findings must be confirmed in clinical trials.

## Supporting information

**S1 File. Additional supporting information may be found in the online version of this article.** Supporting Information file contains values of literature concentrations used in the model and reaction equations and kinetic constants used in the model. The source code is freely available for download at https://github.com/UTP-WTIiE/IrsMtorcQueuesSimulation, implemented in C# supported in Linux or MS Windows.
(DOCX)

## Author Contributions

**Conceptualization:** Sylwester M. Kloska, Tomasz Marciniak, Paul H. Davis, Tadeusz A. Wysocki.

**Data curation:** Marissa Miller.

**Formal analysis:** Tomasz Talaśka.

**Funding acquisition:** Tadeusz A. Wysocki.

**Methodology:** Sylwester M. Kloska, Krzysztof Pałczyński, Tomasz Marciniak, Tadeusz A. Wysocki.

**Project administration:** Tomasz Marciniak, Beata J. Wysocki, Tadeusz A. Wysocki.

**Software:** Krzysztof Pałczyński, Tomasz Talaśka, Marissa Miller.

**Supervision:** Tomasz Marciniak, Tomasz Talaśka, Beata J. Wysocki, Paul H. Davis, Ghada A. Soliman, Tadeusz A. Wysocki.

**Validation:** Sylwester M. Kloska, Krzysztof Pałczyński, Ghada A. Soliman, Tadeusz A. Wysocki.

**Writing – original draft:** Sylwester M. Kloska.

**Writing – review & editing:** Krzysztof Pałczyński, Tomasz Marciniak, Marissa Miller, Beata J. Wysocki, Paul H. Davis, Ghada A. Soliman, Tadeusz A. Wysocki.

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
