## [Decision Letter · Decision Letter 0]

22 Sep 2022

PONE-D-22-19018Queueing theory model of mTOR complexes’ impact on Akt-mediated cell response to insulinPLOS ONE

Dear Dr. Kloska,

Thank you for submitting your manuscript to PLOS ONE. After careful consideration, we feel that it has merit but does not fully meet PLOS ONE’s publication criteria as it currently stands. Therefore, we invite you to submit a revised version of the manuscript that addresses the points raised during the review process. Even after downloading a high resolution of figure 1, it is impossible to read and understand the flow of the signaling. Please also clearly describe, which tissues/cells you are modeling for and why, as the insulin receptor signaling is vastly different in WAT, BAT, liver, muscle, adipose, pancreatic and other cells.  As suggested by the reviewer, please clearly separate introductory background description from the original research performed for this manuscript. Please provide additional clarifications and descriptions, requested by the reviewer. 

We look forward to receiving your revised manuscript.

Kind regards,

Irina U Agoulnik, Ph.D.

Academic Editor

PLOS ONE

Journal Requirements:

“This work was supported by the National Science Center (NCN) of Poland in terms of Opus-17 290 Program [2019/33/B/ST6/00875].”

“This work was supported by the National Science Center (NCN) of Poland (https://www.ncn.gov.pl/) in terms of Opus-17 Program [2019/33/B/ST6/00875 awarded to TAW].

Reviewers' comments:

Reviewer's Responses to Questions

**Comments to the Author**

1. Is the manuscript technically sound, and do the data support the conclusions?

Reviewer #1: Partly

2. Has the statistical analysis been performed appropriately and rigorously? 

Reviewer #1: N/A

3. Have the authors made all data underlying the findings in their manuscript fully available?

Reviewer #1: No

4. Is the manuscript presented in an intelligible fashion and written in standard English?

Reviewer #1: Yes

5. Review Comments to the Author

Reviewer #1: The authors/laboratory have previously described the computational simulation of cellular GLU4 translocation due to insulin secretion. They have also shown other applications of the method, including the initiation of immune response. Here, they expand the previous model to include the interaction of mTOR complexes on PI3K/AKT insulin signalling. However, the authors need to do more to showcase the previous work (not just reference it), while explaining how the present manuscript builds on/is different from what they (and others?) have already published. The authors may choose to present the work as a validation of already-available information via other methods that could potentially yield novel information, but it is important to be clear on what the current goal is, how much of the goal has been achieved in this manuscript and what future perspective the authors envision.

From Line 62 to 104, the authors have did well to explain the historical and current use of “Queueing theory”. The methodology section also attempts to explain the detailed equations, constants, concentrations and algorithms used to compute the model. However, the authors should consider defining the theory itself, and the hypothesis upon which its viability is predicated.

In the methodology and result sections, it is important to be clear about where the authors are showcasing already-available information and where they are discussing their methods/findings, else, they risk presenting a descriptive paper that may not be strongly showcasing their contribution to knowledge.

Summarily, the presentation of the results here is not laid out in a way that a reader can follow in a step wise manner. In modelling specific points in a pathway, it is important for the reader to know what sequence the authors are following in describing their results, especially when it is all tied up in a theory, else the work becomes entirely descriptive and it becomes difficult to differentiate published work from new findings. If it helps, the authors may consider separating the results from the discussion section so that they can explore each section individually and help the reader arrive at a logical conclusion of whether their claims are valid or not.

Minor Points.

Line 92-94: Fig 1 is presented as a low resolution image, making it difficult/impossible to review.

Line 106: It seems intuitive to simple state that “The endpoints of the signaling pathway..”, but the authors may consider stating which pathway they are refereeing to.

Line 198-207: The effect of lowering mTORC1 activity is described here, but the authors should consider leveraging evidences from literature to bolster their findings.

Line 214-220: These points here (as in many parts of the write-up) are available in literature and may not be the results for this manuscript. It is therefore important to appropriately reference them, as this is ethically required and further helps the reader to differentiate between the authors results and evidences in literature (even if those evidences were reported by the authors).

Line 226: “Insulin signaling pathway, disease and treatment strategies”. The authors have majorly discussed mTORC1 activity here, so it is not clear how the subtitle fits the section.

Line 228-229: “The equations described in the model are based on the mass action law”. Please clarify which model is been referred to; the model used by Sonntag et al. or the model used in this study?

Data availability: The authors state that “Yes - all data are fully available without restriction”, is this by request or in a publicly available platform?

6. PLOS authors have the option to publish the peer review history of their article (what does this mean?). If published, this will include your full peer review and any attached files.

Reviewer #1: **Yes: **Olawande C. Olagoke, PhD.

---

## [Author Response · Author response to Decision Letter 0]

28 Sep 2022

Dear Dr Irina U Agoulnik,

Thank you for giving us the opportunity to submit a revised draft of our manuscript titled “Queueing theory model of mTOR complexes’ impact on Akt-mediated cell response to insulin” to PLOS ONE. We appreciate the time and effort that you and the reviewers have dedicated to providing your valuable feedback on my manuscript. We are grateful to the reviewers for their insightful comments on my paper. We have been able to incorporate changes to reflect most of the suggestions provided by the reviewers. We have highlighted the changes within the manuscript.

Here is a point-by-point response to the reviewers’ comments and concerns.

Comments from the Editor

Even after downloading a high resolution of figure 1, it is impossible to read and understand the flow of the signaling. 

Response: Thank you for pointing this out. We agree with your point and we refined mentioned figure in order to improve its quality and readability.

Please also clearly describe, which tissues/cells you are modeling for and why, as the insulin receptor signaling is vastly different in WAT, BAT, liver, muscle, adipose, pancreatic and other cells.

Response: Thank you for pointing this out. As suggested, we have added an explanation that presented model refers to human adipocytes (lines 118-119). 

As suggested by the reviewer, please clearly separate introductory background description from the original research performed for this manuscript. Please provide additional clarifications and descriptions, requested by the reviewer.

Response: Thank you for these suggestions. We have adapted our work to the guidance and feedback provided in the review.

Comments from Reviewer 1

• Comment 1: The authors need to do more to showcase the previous work (not just reference it), while explaining how the present manuscript builds on/is different from what they (and others?) have already published. The authors may choose to present the work as a validation of already-available information via other methods that could potentially yield novel information, but it is important to be clear on what the current goal is, how much of the goal has been achieved in this manuscript and what future perspective the authors envision. However, the authors should consider defining the theory itself, and the hypothesis upon which its viability is predicated.

Response: Thank you for pointing this out. We agree with this comment. Therefore, we have added the following comments in the Introduction section: “The research hypothesis of this work is the ability to simulate the cellular response to insulin and track changes in the concentrations of proteins involved in this response using queueing theory based simulation model. The presented model shows the mechanism of mTORC1 influence on mobilization of GLUT4 particles. Since mTORC1 has been reported in literature as having an impact on glucose uptake (20,21).” 

“Since then the mTORC1 signaling pathway connections with Akt-mediated insulin response has been described (22,23). This work presents a model where those connections have been included.”. 

• Comment 2: In the methodology and result sections, it is important to be clear about where the authors are showcasing already-available information and where they are discussing their methods/findings, else, they risk presenting a descriptive paper that may not be strongly showcasing their contribution to knowledge.

Response: Thank you for this feedback. We have divided the Results and Discussion section into two separate sections, making our results more clearly marked. In this way, the results of the research we conducted are more visible.

• Comment 3: The presentation of the results here is not laid out in a way that a reader can follow in a step wise manner. In modelling specific points in a pathway, it is important for the reader to know what sequence the authors are following in describing their results, especially when it is all tied up in a theory, else the work becomes entirely descriptive and it becomes difficult to differentiate published work from new findings. If it helps, the authors may consider separating the results from the discussion section so that they can explore each section individually and help the reader arrive at a logical conclusion of whether their claims are valid or not.

Response: Thank you for your suggestion. We have separated the Results and Discussion sections so that the visibility of the results obtained from our research is more emphasized.

Minor points:

• Line 92-94: Fig 1 is presented as a low resolution image, making it difficult/impossible to review.

Response: We have revised figures in the paper. We agree with your point and we refined mentioned figure in order to improve its quality and readability.

• Line 106: It seems intuitive to simple state that “The endpoints of the signaling pathway..”, but the authors may consider stating which pathway they are refereeing to.

Response: Thank you for pointing this out. As suggested, we have added an explanation that we meant ‘Akt-mediated insulin signaling pathway’ in this sentence.

• Line 198-207: The effect of lowering mTORC1 activity is described here, but the authors should consider leveraging evidences from literature to bolster their findings.

Response: Thank you for pointing this out. We have added a piece of information in which we refer to literature results that support the results we obtained. “Similar conclusions can be drawn from the results presented by Rajan et al. (39) and Veilleux et al. (40) which confirms the validity of the method we presented.”

• Line 214-220: These points here (as in many parts of the write-up) are available in literature and may not be the results for this manuscript. It is therefore important to appropriately reference them, as this is ethically required and further helps the reader to differentiate between the authors results and evidences in literature (even if those evidences were reported by the authors).

Response: Thank you for raising this issue. We have corrected the indicated fragment to more clearly show the findings of another research team and have cited the work of that team accordingly (lines 226-227).

• Line 226: “Insulin signaling pathway, disease and treatment strategies”. The authors have majorly discussed mTORC1 activity here, so it is not clear how the subtitle fits the section.

Response: Thank you for pointing this out. We have decided to remove subsections, because we divided Results and Discussion sections.

• Line 228-229: “The equations described in the model are based on the mass action law”. Please clarify which model is been referred to; the model used by Sonntag et al. or the model used in this study?

Response: Thank you for pointing this out. We added an explanation which model is been referred to.

• Data availability: The authors state that “Yes - all data are fully available without restriction”, is this by request or in a publicly available platform?

Response: Thank you for your question. The data will be available in the online version of the paper (tables with values of literature concentrations used in the model and reaction equations and kinetic constants used in the model). The source code is freely available for download at https://github.com/UTP-WTIiE/IrsMtorcQueuesSimulation, implemented in C# supported in Linux or MS Windows.

We have also generated a DOI for our GitHub repositorium, which can be accessed here: https://doi.org/10.5281/zenodo.7117138.

We look forward to hearing from you in due time regarding our submission and to respond to any further questions and comments you may have.

Sincerely,

Sylwester Michał Kloska on behalf of all co-authors

---

## [Decision Letter · Decision Letter 1]

28 Nov 2022

PONE-D-22-19018R1Queueing theory model of mTOR complexes’ impact on Akt-mediated cell response to insulinPLOS ONE

Dear Dr. Kloska,

Thank you for submitting your manuscript to PLOS ONE. After careful consideration, we feel that it has merit but does not fully meet PLOS ONE’s publication criteria as it currently stands. Therefore, we invite you to submit a revised version of the manuscript that addresses the points raised during the review process.

Please provide requested references and corrections requested by the reviewer. Please also consider adding a statement to your discussion describing the limitations of your model.

We look forward to receiving your revised manuscript.

Kind regards,

Irina U Agoulnik, Ph.D.

Academic Editor

PLOS ONE

Journal Requirements:

Reviewers' comments:

Reviewer's Responses to Questions

**Comments to the Author**

1. If the authors have adequately addressed your comments raised in a previous round of review and you feel that this manuscript is now acceptable for publication, you may indicate that here to bypass the “Comments to the Author” section, enter your conflict of interest statement in the “Confidential to Editor” section, and submit your "Accept" recommendation.

Reviewer #1: (No Response)

2. Is the manuscript technically sound, and do the data support the conclusions?

Reviewer #1: Yes

3. Has the statistical analysis been performed appropriately and rigorously? 

Reviewer #1: N/A

4. Have the authors made all data underlying the findings in their manuscript fully available?

Reviewer #1: Yes

5. Is the manuscript presented in an intelligible fashion and written in standard English?

Reviewer #1: Yes

6. Review Comments to the Author

Reviewer #1: The authors have done well to respond to the suggestions. They have correctly shown in the methodology that their model and supporting (referenced) experimental result is in adipocytes. They may choose to also let this reflect in the title and/or abstract so as to guide readers to the specificity of their work. Also, appropriate referencing should be made for line 31-36, 63-67 and 121-122. Furthermore, the authors may choose to reword their conclusion as the novelty of their work doesn’t seem to be that “manipulating mTORC1 kinase affects GLUT 4 translocation and glucose entry”, but that queuing theory can effectively model the process.

Importantly, the authors have alluded to the improvement of queuing theory over ODE’s. are there any drawbacks to modelling with the theory, especially in relation to mTORC1 mediated GLUT 4 translocation?

7. PLOS authors have the option to publish the peer review history of their article (what does this mean?). If published, this will include your full peer review and any attached files.

Reviewer #1: **Yes: **Olawande Olagoke, PhD.

---

## [Author Response · Author response to Decision Letter 1]

29 Nov 2022

Dear Dr Irina U Agoulnik,

Thank you for giving us the opportunity to submit a revised draft of our manuscript titled “Queueing theory model of mTOR complexes’ impact on Akt-mediated adipocytes response to insulin” to PLOS ONE. We appreciate the time and effort that you and the reviewers have dedicated to providing your valuable feedback on my manuscript. We are grateful to the reviewers for their insightful comments on my paper. We have been able to incorporate changes to reflect most of the suggestions provided by the reviewers. We have highlighted the changes within the manuscript.

Here is a point-by-point response to the reviewers’ comments and concerns.

Comments from the Editor

Response: Thank you for these notifications. We have checked references and we believe there is no retracted paper what we refer to. We have made every effort to ensure that the list of references is complete and meets the requirements of the Journal.

Comments from Reviewer 1

• The authors have done well to respond to the suggestions. They have correctly shown in the methodology that their model and supporting (referenced) experimental result is in adipocytes. They may choose to also let this reflect in the title and/or abstract so as to guide readers to the specificity of their work. 

Response: Thank you for your suggestions. We have changed the manuscript title and abstract. They now clearly show that the paper is related to adipocytes response to insulin.

• Also, appropriate referencing should be made for line 31-36, 63-67 and 121-122. 

Response: Thank you for pointing this out. We have added new references in the mentioned lines.

• Furthermore, the authors may choose to reword their conclusion as the novelty of their work doesn’t seem to be that “manipulating mTORC1 kinase affects GLUT 4 translocation and glucose entry”, but that queuing theory can effectively model the process.

Response: Thank you for pointing this out. We agree with the comment. The relevant statement has been included in the Conclusions section.

• Importantly, the authors have alluded to the improvement of queuing theory over ODE’s. are there any drawbacks to modelling with the theory, especially in relation to mTORC1 mediated GLUT 4 translocation?

Response: The presented model has several limitations. It does not take into account other signaling pathways or individual reactions that are also connected to and influence signaling proteins. This is especially true for the Akt protein, which is the central node in the presented signaling model. Moreover, a model based on literature data will only be as good as the available data. However, we do not question the reliability of other research teams and their published results. Another of the limitations is that in queueing theory, each simulation gives one realization of the stochastic process, while ODE gives an averaged solution. Therefore, a limitation is that depending on the number of cells for which one runs simulations and then averages them, this is how accurate the result will be. Therefore, the model presented here is for averaged results for 50 cells.

We look forward to hearing from you in due time regarding our submission and to respond to any further questions and comments you may have.

Sincerely,

Sylwester Michał Kloska on behalf of all co-authors

---

## [Decision Letter · Decision Letter 2]

12 Dec 2022

Queueing theory model of mTOR complexes’ impact on Akt-mediated adipocytes response to insulin

PONE-D-22-19018R2

Dear Dr. Kloska,

We’re pleased to inform you that your manuscript has been judged scientifically suitable for publication and will be formally accepted for publication once it meets all outstanding technical requirements.

Kind regards,

Irina U Agoulnik, Ph.D.

Academic Editor

PLOS ONE

Additional Editor Comments (optional):

Reviewers' comments:

Reviewer's Responses to Questions

**Comments to the Author**

1. If the authors have adequately addressed your comments raised in a previous round of review and you feel that this manuscript is now acceptable for publication, you may indicate that here to bypass the “Comments to the Author” section, enter your conflict of interest statement in the “Confidential to Editor” section, and submit your "Accept" recommendation.

Reviewer #1: All comments have been addressed

2. Is the manuscript technically sound, and do the data support the conclusions?

Reviewer #1: (No Response)

3. Has the statistical analysis been performed appropriately and rigorously? 

Reviewer #1: (No Response)

4. Have the authors made all data underlying the findings in their manuscript fully available?

Reviewer #1: (No Response)

5. Is the manuscript presented in an intelligible fashion and written in standard English?

Reviewer #1: (No Response)

6. Review Comments to the Author

Reviewer #1: (No Response)

7. PLOS authors have the option to publish the peer review history of their article (what does this mean?). If published, this will include your full peer review and any attached files.

Reviewer #1: **Yes: **Olawande Olagoke, PhD.
